# A novel RP-HPLC method for quantification of cholinesterase activity in human blood: An application for assessing organophosphate and carbamate insecticide exposure

Sukesh Narayan Sinha[1¤a]*, Ramakrishna Ungarala[1¤a], Dileshwar Kumar[1¤a], Rajendra Sangaraju[1¤a], Sarita Kumar[2¤b]

1 ICMR- National Institute of Nutrition, Hyderabad, Telangana, India, 2 Acharya Narendra Dev College, University of Delhi, New Delhi, India

¤a Current address: Food Safety Division, National Institute of Nutrition-ICMR, Hyderabad, Telangana, India
¤b Current address: Department of Zoology, Acharya Narendra Dev College, University of Delhi, Kalkaji, New Delhi, India
* sukeshnr_sinha@yahoo.com

**Data Availability Statement:** All relevant data are within the paper.

## Abstract

Several methods have been reported to estimate Acetylcholinesterase (AChE) enzyme activity in blood samples. The Ellman assay is the most important among all but with several shortcomings, and there is a need to develop a method which is accurate, sensitive and quick for analyzing AChE. Therefore, we have developed an assay utilizing RP-HPLC with UV detection for the determination of AChE activity. This method measured the conversion of 1-naphthol acetate to 1-naphthol to estimate AChE activity in blood samples. Performance was judged on the basis of reproducibility, sensitivity, accuracy, and the ability to screen enzyme activity within 20minutes. A series of experiments were performed, varying the concentration of blood and substrate, with optimal sensitivity using 50 µM substrate and 10µL blood. The validation parameters such as linearity ($R^2$ of $\geq$ 0.9842 for 1-naphthol and $\geq$ 0.9897 for 1-naphthol acetate), precision (94.21–96.41%), accuracy (85.2%–99.6% and 82.6%–99.3% for 1-naphthol and 1-naphthol acetate respectively), and robustness were validated according to International Conference on Harmonization (ICH) guidelines. Blood samples were collected from healthy people, farmers exposed to spraying of pesticides, and suicidal patients who ingested pesticides and were hospitalized and were analyzed by the developed method. The AChE level was approximately 21 units/mL compared to 24units/mL in controls, whereas suicidal patients showed the least AChE levels of 1 unit/mL. The employment of this method is recommended for estimating AChE level on various matrices.

## Introduction

Pesticides are routinely used in agriculture, especially organophosphorus (OP) and carbamate pesticides. Pesticide residues in drinking water and food have sparked health concerns. There

**Funding:** The study has been funded by the Department of Health Research (F.No. V. 25011/539-HRD/2016-HR, Ministry of Health and Family Welfare, Govt. of India. The funders had no role in study design, data collection, and analysis, decision to publish, or preparation of the manuscript.

**Competing interests:** The authors have declared that no competing interests exist.

is strong evidence that acute and chronic exposure to these chemicals harms the neurological system. The major enzyme that breaks down the neurotransmitter acetylcholine (ACh) in the nervous system, acetylcholinesterase (AChE), is crucial to neuromuscular and brain function [1]. Inhibition of AChE activity has long been utilized as a biomarker for organophosphorus pesticide exposure.

Synaptic accumulation of neurotransmitters occurs when nerve agents inhibit the enzymatic activity of acetylcholinesterase (AChE) in the nervous system. Nerve agents are harmful and are absorbed through the skin and lungs. In battle, nerve poisons and anticholinesterase chemicals can paralyse or kill. Community preparedness and military locations that produce or store agents need reliable reports on agent properties and treatment and clear health-based vulnerability standards based on contemporary data analysis [2, 3].

The measurement of erythrocyte AChE activity is the best way to diagnose OP's exposure and intoxication [4, 5] developed the first potentiometric method for determining AChE activity by measuring pH for 1 hour. The Ellman photometric method uses the rise in yellow hue caused by thiocholine whet it combining with dithiobisnitrobenzoate (DTNB) ion to measure acetylcholinesterase activity in tissue extracts, homogenates, cell suspensions, and other substances [6, 7].

The Michel technique and Ellman photometry yielded comparable AChE activity results. Still, the colorimetric approach was shown to be more accurate [8]. Several blood sample AChE activity techniques have been reported [9–16]. However, buffers, pH fluctuation, blood in microliters, and time are drawbacks of published procedures [17, 18] examined substrate concentration, pH, and temperature using titrimetry [19]. Since its inception, titration has been employed to measure AChE activity [20, 21]. This method like the electrometric method, involves unbuffered and buffered titrations.

A method for estimating AChE activity using the fluorescence of resorufin produced by enzyme interactions involving acetylcholine and Amplex Red was recently reported [22]. Although mass spectrometric approaches for estimating enzyme activity in biological samples have been published [23–25], these methods are expensive, sophisticated, and time-intensive.

In medicine, toxicology, and civil protection, quantifying AChE activity is a useful technique [7, 26–28]. In the realm of AChE determination, the Ellman photometric approach is still the gold standard [6, 7]. However, the Ellman photometric approach is not without flaws [29, 30]. Several approaches based on a version of the Ellman method have been reported [31–33], but none displays total enzymatic volume.

As a result, this study's focus is investigating 1-naphthol acetate as a potential substrate for AChE activity detection at pH 7.0 using the HPLC technique coupled with a PDA-based detection system operating at 280 nm. Using 1-naphthol acetate as a substrate instead of acetylthiocholine is expected to be a reliable way to measure cholinesterase activity, avoiding the drawbacks of acetylthiocholine [34, 35]. Furthermore, 1-NA may be a more appealing chromogenic substrate for measuring AChE activity than Acetylthiocholine or Butyryl thiocholine. In terms of lower Km value, 1-NA appears to be a better alternative substrate for AChE activity than acetylthiocholine (ATCh). Its specificity appeared at least similar to ATCh. Therefore, the authors proposed that 1-NA can be an attractive chromogenic substrate for the measurement of AChE activity, and it possesses the potential to detect organophosphorus pesticide (OP) poisoning [34].

Hence, it necessitated the need for a well-developed and validated method for the identification, early detection and estimation of AChE, in human biological samples, due to intended or unintended exposures of poisoning conditions and chemical warfare conditions. Therefore, we aimed to develop techniques to observe AChE inhibition with sufficient sensitivity and could be performed using very few μL of human blood.

## Materials and methods

The Osmania General Hospital Ethics Committee (Reg No. E.C.R./300/Inst/A.P./2013) authorized experimental protocols to collect blood samples from healthy and poison-exposed subjects to conduct the present study. All the methods were performed as per the standard ethical guidelines and regulations. Informed consent was taken from all the subjects/ guardians before the collection of the blood samples.

### Chemicals

HPLC-grade solvents and analytical-grade (AR) reagents were utilized for the study. HPLC-grade acetonitrile, methanol solvents, and water were purchased from M/s J.T. Baker Avanator (Radnor, Pennsylvania, U.S.A.). 1-naphthol and1-naphthol acetate were procured from Sigma (St. Louis, MO, U.S.A.). Sodium hydrogen phosphate buffer (pH 6.88) was procured from Merck, Germany. AChE (human) was procured from Sigma-Aldrich with CAS number of 9000-81-1 and a product number of C0663. The same was used for the reaction in our manuscript.

### Stock and working solution

For the study, standard stock solutions of 1 M for 1-naphthol and1-naphtholacetate were prepared. Six working standard sets were prepared for 1-naphthol and1-naphthol acetate at concentrations of 50, 60, 70, 80, 90 and 100 μM in acetonitrile for validation of method. These standards were then used for the analysis and reaction kinetics study experiment.

**Collection of blood and isolation of RBC.** The 2 ml whole blood was collected from healthy volunteers and then subjected to a reaction without freezing the samples. A sample volume of 10 μL from the 2 ml collected blood was used for the reaction. The blood samples were collected from venous blood and placed in test tubes. Capillaries and test tubes should be heparinized (to prevent blood clotting) and dried (to prevent uncontrolled sample dilution). To prevent contamination of the samples by OPs and carbamates during collection, the skin must be cleaned before sampling. Similarly, The RBC were separated from whole blood by centrifuging it at 500 x g for 10 min at 4 degrees C. Aspirate the supernatant (plasma) and add cell wash buffer to the erythrocyte pellet. Discard the supernatant and repeat washing with 2 ml of PBS (Phosphate buffered saline) in order to remove the residual plasma and any residual OPs or carbamates that may be present in the blood [36].

### Instrumentation

The 1-naphthol acetate and 1-naphthol were analyzed using Shimadzu 20AD dual pumps, and auto-injector, a PDA detector (Shimadzu, Japan) installed with an LC Solutions Software and printer fitted with a $C_{18}$ reverse phase column and operated in isocratic solvent mode.

**Liquid chromatography conditions (L.C.).** All substances were separated chromatographically using a liquid chromatography with a C18 reversed-phase column (150 x 4.6 mm ID, 4.5 μ particle size). The Shimadzu auto-sampler was used to inject samples (20 μL) into the device. Then a 20μL reaction mixture was injected into the RP-HPLC. All measurements were carried out at room temperature. The isocratic composition of mobile phase water–acetonitrile (55: 45, v/v) pumped at a flow rate of 1 mL/min provided the best sensitivity and separation of naphthol compounds. All the standards of 1-naphthyl acetate were prepared in acetonitrile. The temperature in the column oven was kept constant at 25 ˚C. At a wavelength of 280 nm, the absorbance was measured. and the data were integrated into the software. Generally, the enzyme reaction takes place at pH 7 in the buffer solution. Therefore, the phosphate buffer was

used to carry out the reaction in blood samples, and acetonitrile was used to stop the reaction to optimize the kinetics of the reaction at different time periods. while phosphate buffer (pH 6.88) medium was used to carry out the reaction in blood samples and acetonitrile was used for stopped the reaction in blood samples.

## Method validation

During the method validations AChE or blood were not used.

**Specificity.** It is essential to check the proper separation of analyte peaks from any other interfering peaks that may have arisen from matrix of the sample [37–39]. By introducing 20 μL of buffer solution as placebo and the mobile phase as a blank, we tested and evaluated the procedure in the RP-HPLC system.

**Linearity.** The developed method's linearity was tested by diluting stock solutions with the mobile phase and injecting various concentrations of sample solutions, such as 50, 60, 70, 80, 90, and 100 μM before the mobile phase was analyzed on a RP-HPLC system.

**Accuracy.** To determine the accuracy, the solutions were spiked with reference standards of 1-naphthol and 1-naphthol acetate in different concentrations (low, mid, and high) and were injected as different runs into the buffer matrices at concentrations of 50%, 80%, and 100%. For each concentration, triplicate aliquots of solutions were produced and tested, and the recovery for both substances was determined.

**System suitability.** When a method is subjected to multiple homogeneous samples, precision refers to the consistency of the test results. The 80 μM standard solutions of 1-naphthol and 1-naphthol acetate were injected six times for evaluation of system suitability.

**Limit of detection.** The least detectable limit (DL) of analyte in a test sample is known as the limit of detection (L.O.D.). Eq 1 was used to compute the L.O.D.

$$LOD = (3.3\ \sigma)/S \qquad\qquad Eq\ (1)$$

where, σ = SD obtained from responses

S = slope obtained from the calibration curve.

**Limit of quantification.** This quantitative limit (Q.L.) of the lowest concentration or limit of quantification (L.O.Q.) is generally 10 times the signal-to-noise ration. Eq 2 was used to determine the L.O.Q.

$$LOQ = (10\ \sigma)/S \qquad\qquad Eq\ (2)$$

where, σ = standard deviation obtained from responses

S = slope obtained from calibration curve

**Robustness.** The analytical method's robustness was tested by adjusting the flow rate of mobile phase modifications in the analytical process. For each condition, triplicate solutions of 1-naphthol and 1-naphthol acetate were introduced into the system. After both substances solutions were injected three times, the percent RSD of the peak area was calculated, and the method's reliability was determined.

**Precision.** Precision was tested by injecting six distinct aliquots of 1-naphthol and 1-naphthol acetate, which were utilized to test the ruggedness of the procedure for this validation.

## Auto hydrolysis of naphthyl acetate in the absence of AChE

To confirm that AChE is involved in the conversion of 1-naphthol acetate to 1-naphthol, the following assay was performed. 50 μL clean water (LC-MS grade) add with 50 μL solution of 50 μM of 1-naphthol acetate was added and followed by AChE (2 unit/μL) in 200μL phosphate

buffer (pH 6.88). After different time intervals, the reaction was stopped using with 700 μL acetonitrile and filtered using a 0.2μ syringe filter and collect the filtrate of 20μL reaction mixture was injected into RP-HPLC under the same conditions mentioned above. The peak of 1 naphthol was obtained in the presence of AChE, while in absence of AChE in similar condition (in water) no peak of 1 naphthol was obtained. This experiment was performed to check autohydrolysis of 1-naphthol acetate in the presence of water.

## Assay optimization in blood

Blood samples in buffers along with the substrate (1-naphthol acetate) were incubated at room temperature for 20 minutes, after which the addition of acetonitrile stopped the reaction and the samples were filtered with 0.2μ syringe filters, and the filtrate was injected into the RP-HPLC system. The conversion of 1-naphthol acetate to 1-naphthol catalysed by the AChE in the blood was followed by the simultaneous disappearance of the former and the appearance of the latter on the chromatograms.

Different concentrations and blood volumes were tried with a 50 μM concentration of 1-naphthol acetate that was standardised for the reaction and estimation. Finally, 10 μL of blood was standardised for the estimation. Following that, 10 μL of blood was diluted in 280 μL of phosphate buffer and mixed with 10 μL of 50 μM 1-naphthol acetate. The reaction was carried out at room temperature and up to 20 minutes, the reaction was stopped using with 700 μL acetonitrile. Next, 20 μL filtrate was injected to RP-HPLC system. The complete conversion of 1-naphthol acetate to 1-naphthol was achieved using a 50 μM concentration in the acetonitrile reaction mixture within 20 minutes of the reaction initiation, which proves that the enzyme is very reactive and responsible for the conversion of 1-naphthol acetate to 1-naphthol. Similarly, 20 μL of phosphate buffer was injected on RP-HPLC, whereas no peak was observed, which acts as a control. This procedure is depicted in the schematic below Fig 1.

**Assay optimization in RBC samples.** Erythrocyte samples and the substrate (1-naphthol acetate) were put in a buffer at room temperature for 20 minutes as described in above method and shown in (Fig 1). The reaction was stopped when acetonitrile was added, and the samples were filtered with 0.2-micron syringe filter paper. The filtrate was then put into the RP-HPLC system. The conversion of 1-naphthol acetate to 1-naphthol, catalysed by the AChE present in

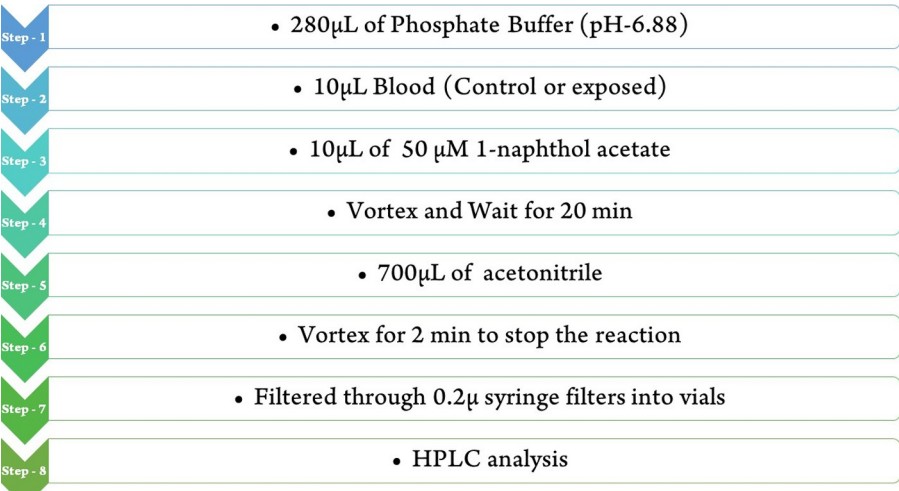

| | |
|---|---|
| Step - 1 | • 280μL of Phosphate Buffer (pH-6.88) |
| Step - 2 | • 10μL Blood (Control or exposed) |
| Step - 3 | • 10μL of 50 μM 1-naphthol acetate |
| Step - 4 | • Vortex and Wait for 20 min |
| Step - 5 | • 700μL of acetonitrile |
| Step - 6 | • Vortex for 2 min to stop the reaction |
| Step - 7 | • Filtered through 0.2μ syringe filters into vials |
| Step - 8 | • HPLC analysis |

**Fig 1. Schematic representation of reaction procedure.**

the RBC membrane, was followed by the simultaneous disappearance of the former and the appearance of the latter on the chromatograms. Several concentrations and blood volumes were tried, and a 50 µM concentration of 1-naphthol acetate was standardised for the reaction and estimation.

### AChE estimation in blood samples of exposed and control cases

2 ml of blood was collected into EDTA tubes from healthy people (n = 10), farmers (n = 78) exposed to pesticides while spraying, and suicidal patients (n = 3) hospitalized after ingesting pesticides. All the blood samples were subjected to the reaction procedure and analyzed using the developed method, and AChE levels were estimated. Following that, 10 µL of blood was diluted in 280 µL of phosphate buffer and mixed with 10 µL of 50 µM 1-naphthol acetate. The reaction was carried out at room temperature and up to 20 minutes, the reaction was stopped using with 700 µL acetonitrile. The acetonitrile was used to stop the reaction to inhibit AChE enzyme activity on RBC, and thereafter the reaction mixture was filtered using 0.2µ syringe filters. The 20µL filtrate was injected into the RP-HPLC system. The AChE in the blood catalysis the conversion of 1-naphthol acetate to 1-naphthol, and the data is integrated with the chromatogram. The enzyme activity is measured in the units which indicate the rate of the reaction catalyzed by that enzyme expressed as micromoles of substrate transformed (or product formed) per minute. The enzyme activity was expressed as U/mL.

### Result

The physicochemical properties of 1-naphthol and 1-naphthol acetate used to evaluate the ruggedness by different methods of acetylcholinesterase on different matrices were obtained from the literature Table 1. Attempts to separate the reaction products using polar and nonpolar HPLC solvents such as methanol, ethyl acetate, water, and acetonitrile, as well as combinations of these solvents, failed. It may be noted that enzymatic hydrolysis behaved entirely differently in basic and acidic mediums and, therefore acidic and basic conditions were not used for the reaction. The reaction was carried out in neutral pH 7 (sodium phosphate buffer).

The compounds were dissolved at various concentrations in acetonitrile and the absorbance was measured. The spectra were acquired, and it was discovered that at 280 nm, both 1-naphthol and 1-naphthol acetate have a significant absorption.

It was found that the separation was easily achieved using acetonitrile–water (45:55, v/v) as mobile phase on $C_{18}$ reversed-phase column with a flow rate of 1.0 mL/min. Fig 2 shows the HPLC chromatogram of the standard mixture of 1-naphthol (R.T.: 4.7 min) and1-napthola- cetate (R.T.: 8.2min) at concentrations ranging from 50–100 µM. The running time of the

**Table 1. Literature review of HPLC methods for assaying AChE activity and inhibition.**

| S. No. | Mobile Phase | Flow Rate / Run time | Fluorescence / Absorbance | Detection | Reference |
|---|---|---|---|---|---|
| 1 | Methanol: Water: Diethyl Amine (40:60:0.05) | 1.2 mL/min | 405 nm | AChE inhibitor, galantamine | [40] |
| 2 | Methanol: Water (48:52) | 0.8 mL/min, 12 min | 227 nm, 355 nm | 2-naphthyl acetate to form 2-naphthol | [40] |
| 3 | Methanol: Water: Triethylamine (40:60:0.05) | 1.0 mL/min 5 min | 405 nm | AChE inhibitory activity using galantamine hydrobromide and huperzine | [41] |
| 4 | Acetonitrile: Water: Orthophosphoric acid (85%) (10:90:0.1) | 0.4 mL/min | 273 nm | pyridin-2-ylmethyl acetate as substrate for AChE | [42] |
| 5 | Methanol: Water 25: 75 | 0.2 mL/min | LC-MS/MS | Electrophorus electrics acetylcholinesterase (Eel Ache) inhibition | [43] |

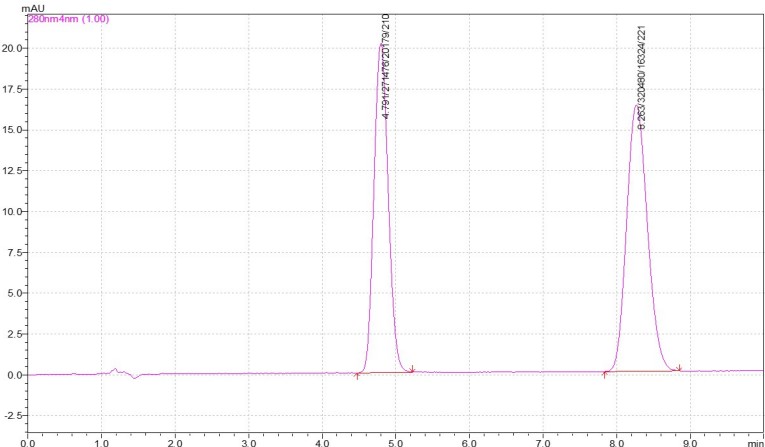

**Fig 2. Chromatogram of 1-naphthol and 1-naphthol acetate on the developed method.**

chromatogram was 10 min. The change of retention time (R.T.) may be ± 0.1 min for each run but the relative retention time was the same for all compounds throughout the analysis.

In addition, the current study was time-saving and simple because it followed an isocratic flow and produced better elution results than associated with a time-consuming gradient flow, which demonstrated the method's rapidity and cost-effectiveness because it only requires a limited amount of mobile phase.

## Method validation

**Specificity.** Analyses of blank and placebo solutions show that the technique devised is very specific for recognizing 1-naphthol and 1-naphthol acetate, and it is in accordance with the specifications of the International Conference on Harmonization (ICH) (Fig 3).

**Linearity.** At linear concentration solutions of 50, 60, 70, 80, 90, and 100 μM, 1-naphthol and 1-naphthol acetate were injected in triplicate, and analyte peak areas were observed. Fig 4 illustrates that a plot was produced for both chemicals against various concentrations.

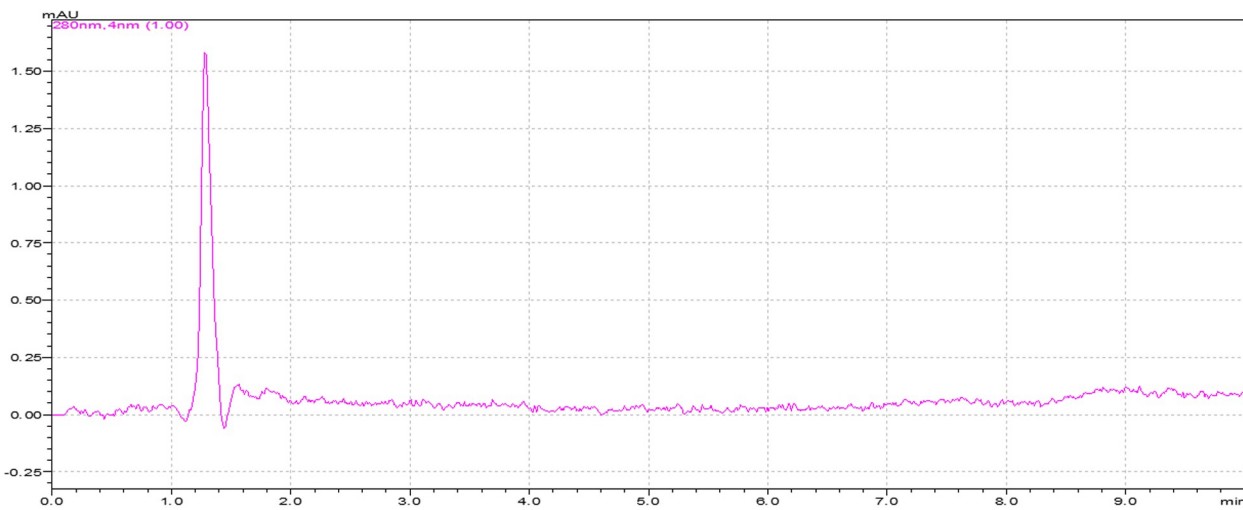

**Fig 3. Chromatogram of placebo on the developed method.**

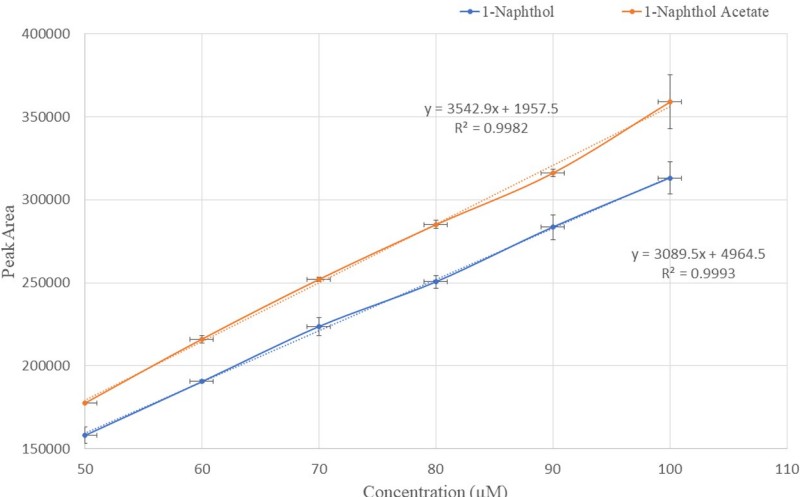

**Fig 4. Regression analysis of linear concentrations of 1-naphthol and 1-naphthol acetate.**

Regression analysis was employed, and $R^2$ was observed to be $\geq$ 0.9993 for 1-naphthol and $\geq$ 0.9982 for 1-naphthol acetate. The linear equations were y = 3089.5x + 4964.5 and y = 3542.9 + 1957.5 for 1-naphthol and 1-naphthol acetate, respectively (Fig 4).

**Accuracy.** The optimized method's recovery studies were carried out, and the percent recovery was calculated Table 2. At all three concentration levels, the recovery rates were obtained in the range of 85.2%–99.6% and 82.6%–99.3% for α-naphthol and α-naphthol acetate.

**System suitability.** A concentration of 80 M of 1-naphthol and 1-naphthol acetate is used to test the analytical system for repeatability, accuracy, and precision. For 1-naphthol and 1-naphthol acetate, the percent RSD of RT was 0.81 and 0.93, respectively, while the percent RSD of peak area was 0.7 and 0.63.

**L.O.D. and L.O.Q.** The signal-to-noise ratios of 3.3 and 10 were used to determine LOD and LOQ, respectively. For 1-naphthol and 1-naphthol acetate, the standard deviations (σ) were 1786.61 and 1066.73, respectively. The calibration curve's slope (S) from linearity was 2866.4 and 3588.4, respectively. As a result, the L.O.D.s for 1-naphthol and 1-naphthol acetate

**Table 2. Recoveries of 1-naphthol and 1-naphthol acetate on the developed method.**

| Spiked weight of standard (µM) | Replicate number | Concentration of the solution (µg/mL) | Area | | Spiked Area | | Recovery (%) | |
|---|---|---|---|---|---|---|---|---|
| | | | 1-Naphthol | 1-Naphthol acetate | 1-Naphthol | 1-Naphthol acetate | 1-Naphthol | 1-Naphthol acetate |
| 60 | 1 | 60 | 9380 | 10763 | 17616 | 19658 | 87.8 | 82.6 |
| | 2 | 60 | 9495 | 10666 | 17582 | 19696 | 85.2 | 84.7 |
| | 3 | 60 | 9260 | 10820 | 17662 | 19712 | 90.7 | 82.2 |
| 80 | 1 | 80 | 12404 | 14897 | 24363 | 26985 | 96.4 | 81.1 |
| | 2 | 80 | 12367 | 14787 | 24281 | 26971 | 96.3 | 82.4 |
| | 3 | 80 | 12422 | 14812 | 24411 | 26891 | 96.5 | 81.5 |
| 100 | 1 | 100 | 14942 | 17671 | 29706 | 35214 | 98.8 | 99.3 |
| | 2 | 100 | 14881 | 17741 | 29651 | 35192 | 99.3 | 98.4 |
| | 3 | 100 | 14852 | 17701 | 29643 | 35245 | 99.6 | 99.1 |

were determined to be 2.0568μM and 0.9810μM, respectively, while the L.O.Q.s were 6.232μM and 2.972Mm.

**Robustness.** The robustness of the developed method was validated Table 3, shows the results of the flow rate modifications that were made on purpose. The minor alterations made for the robustness test did not result in any significant variations, and the parameters were found to be within ICH's acceptable ranges. No AChE or blood were used during the analysis.

**Precision.** The precision analysis results were calculated, and the percent RSD for 1-naphthol and 1-naphthol acetate was found to be 0.70 percent and 0.38 percent, respectively. The percent assay resulted in a range of 94.21–96.41 percent, which is within the ICH recommendations.

## Assay optimization in blood

To check the enzyme activity buffer solutions with 1-naphthol and 1-naphthol acetate were injected separately into the HPLC system and no conversion of substrates was observed. We have also used phosphate buffer with and without AChE to observe the 1-napthol acetate conversion to 1-napthol which clearly indicates that 1-napthol acetate is being converted to 1-napthol by acetylcholinesterase. Further, blood samples from people of our institute who are not exposed to any pesticide sprays were used for observing the reaction and impact of AChE in conversion of the substrate. The peak areas of the substrate during conversion have been represented in the Fig 5. The curves show the conversion of 1-naphthol acetate to 1-naphthol at different time points. The concentration of 1-naphthol conversion in blood at different time intervals in represented in Fig 6.

## AChE estimation in blood samples

The developed method was applied to blood samples collected from individuals who were not exposed to pesticides by spraying, handling, or ingestion; these samples served as controls. Those directly involved in applying pesticides and suicidal patients provide the exposed samples. The AChE levels in the human samples are represented in the Table 4 below. The farmers

**Table 3. Robustness of 1-naphthol and 1-naphthol acetate on the developed method.**

| Flow Rate (mL/min) | 1-naphthol RT | 1-naphthol acetate RT | 1-naphthol Peak Area | 1-naphthol acetate Peak Area |
|---|---|---|---|---|
| **0.9** | 5.255 | 9.078 | 287929 | 322419 |
| | 5.271 | 9.031 | 282096 | 316576 |
| | 5.258 | 9.078 | 284972 | 319472 |
| | 5.257 | 9.009 | 287213 | 321713 |
| | 5.248 | 9.064 | 287328 | 321818 |
| **Mean** | 5.26 | 9.05 | 285907.60 | 320399.60 |
| **SD** | 0.008 | 0.031 | 2408.906 | 2413.256 |
| **RSD (%) (n = 3)** | 0.16 | 0.34 | 0.84 | 0.75 |
| **1.1** | 4.266 | 7.371 | 231716 | 272346 |
| | 4.288 | 7.403 | 234602 | 275232 |
| | 4.293 | 7.417 | 236833 | 277463 |
| | 4.3 | 7.431 | 236948 | 277578 |
| | 4.302 | 7.436 | 236948 | 277578 |
| **Mean** | 4.2898 | 7.4116 | 235409.4 | 276039.4 |
| **SD** | 0.01 | 0.03 | 2294.25 | 2294.25 |
| **RSD (%) (n = 3)** | 0.34 | 0.35 | 0.97 | 0.83 |

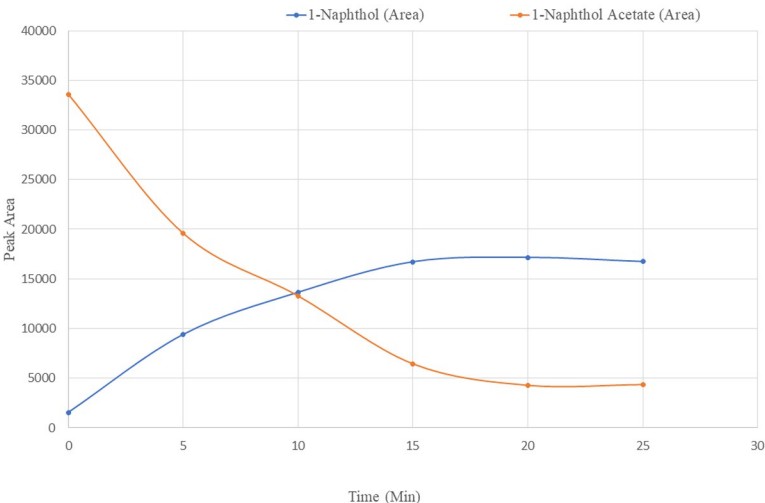

**Fig 5. 1-naphthol acetate conversion to 1-naphthol by AChE in blood at different time intervals.**

who were directly involved in the spraying of pesticides showed AChE mean of 21.84 Unit/mL and SD of±2.33as compared to control samples without any exposure to spraying of pesticides depicted AChE mean of 24.079 unit/mL with SD ± 1.22. The patients (suicidal individuals who had intendedly ingested pesticides and were admitted to hospitals) has shown very low levels of AChE 1.09 unit/mL with SD of ± 1.02.

Further, exposed samples collected from farmers were checked with the pesticide [44, 45] found and compared the acetylcholinesterase levels in the samples and the levels are given in the Table 5 below. The level of ChE in all exposed samples, particularly the pesticide found, was around 22 units/mL, which is comparatively lower than the control subjects.

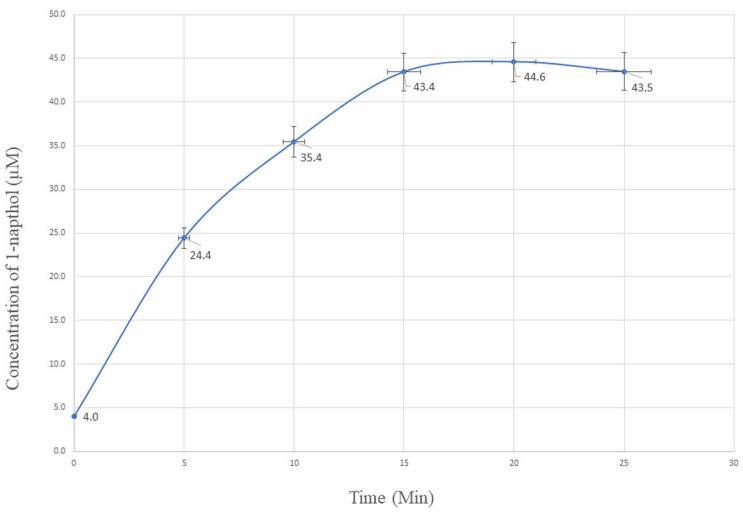

**Fig 6. Concentration of 1-naphthol formed by AChE in blood at different time intervals.**

Table 4. AChE levels estimated in human blood samples of control and exposed.

| Samples | AChE (Unit/mL) |
|---|---|
| Control Samples (n = 10) | 24.1 ± 1.2 |
| Exposed Samples (Farmers) | 21.9 ± 2.3 |
| Exposed Samples (Suicidal/Ingested) | 1.1 ± 1 |

## Discussion

Organophosphate poisoning is a universal challenge. Inhibition of acetylcholinesterase (AChE) is a reliable marker for O.P., carbamate insecticides, metals, and nerve agents poisoning. Thus, the objective of this paper was to develop prototype methods for measuring insecticide exposure and effect which are crucially important to the prognosis within a limited period of time.

The currently available technique used for these determinations is based on the Elman's method where the person's enzyme hydrolyses acetylthiocholine in their blood to thiocholine. The Ellman method was considered as a gold standard method which was a photometric method to determine the acetylcholinesterase activity in different matrices [6]. This method detected enzymatic activity by observing colour change. The 10 μL amount of blood used in our present developed method while Ellman method also utilizes 10μL of whole blood sample. However, these days the most applied technique to determine cholinesterase activity is Ellman's colorimetry. It is suitable for the assessment of biological samples, for the identification of AChE and the determination of their efficacy. But its limitation lies in strong interaction with haemoglobin during whole blood sample analysis. Furthermore, Ellman's colorimetry cannot be uncritically used to assay the oxime reactivators. It may not be used in various protocols for the determination of pesticide or nerve agent exposure. This method used only for haemoglobin-rich samples. It evaluates biological material, identifies AChE, and determines efficacy. During whole blood sample analysis, it interacts heavily with hemoglobin. The Ellman method is hindered by hemoglobin absorption at 412 nm while our HPLC method utilized 280 nm.

Later, a method based on hydrolysis of propionyl thiocholine by spectrophotometric determination of thiocholine was developed by Augustinsson KB et al in 1978. This method was specific to propionyl thiocholine while our method is more specific to cholinesterase which edges over the developed method and also our method was developed on HPLC which is more confirmatory over spectrophotometric method [46]. Johnson and Russel reported radiometric assay is a practical approach for protocols where high sensitivity is needed. Fluorometric methods are useful for sensitive determination within kinetic studies at very low enzyme

Table 5. AChE levels in comparison with different pesticide exposures.

| S. No. | Pesticide | No of Samples | AChE (Unit/mL) |
|---|---|---|---|
| 1. | Acephate | 1 | 23.7 |
| 2. | Acetochlor | 8 | 22.8 ± 1.4 |
| 3. | Butachlor | 62 | 21.9 ± 2.6 |
| 4. | Coumaphos | 45 | 22.1 ± 1.3 |
| 5. | Metribuzin | 1 | 23.5 |
| 6. | Monnocron | 1 | 21.3 |
| 7. | Monocrotophos | 29 | 21.9 ± 1.5 |
| 8. | Profenophos | 2 | 22.9 ± 0.6 |
| 9. | Triazophos | 14 | 20.5 ± 4.6 |

concentrations. Similarly, Biosensors seem to be suitable for OP and carbamate identification. The AChE activity determination approach needs to fulfil several requirements like sufficiently sensitive, selectivity for AChE activity is necessary. Our method is capable for the assessment of biological samples as well as for kinetic studies and new drug discovery at very low level enzyme. Earliest techniques were developed mainly for overall AChE activity determination without accuracy. Great potential lies in use of our methods, which creates new possibilities in this field.

The principles that developed method here could be used in clinical settings as enzyme activities that are impaired by specific exposures are discovered. The in-silico approach by Chowdhary S et al., has also focused on 1-napthyl acetate as a substrate for the estimation of cholinesterase by molecular simulations while we employed the method on real time with samples collected from agricultural farm workers and hospitals by using a volume of a finger prick [34].

Reversed-phase HPLC was used to separate and identify 1-naphthol as a biomarker for carbamate, OP insecticides, and nerve agents. Isocratic separation employed water–acetonitrile (55:45). This solvent mixture is simple; no bi- or poly-gradient proportions were needed. Our method for separating compounds is straightforward. The sensitive HPLC approach estimates AChE in 10 μL of human blood in 20 minutes. According to reports, an HPLC-based assay should be 10 to 1000 times more sensitive than a colorimetric approach (Ellman Method), permitting the creation of assays based on very small blood volumes. Our investigation demonstrated that reversed-phase HPLC was the suitable analytical approach for estimating 1-naphthol, a biomarker for OP pesticides, carbamate insecticides, and nerve agent rather.

Our method can be utilized to investigate biological samples, kinetics, and novel drugs. Early strategies were devised to estimate AChE activity. All approaches utilized natural or recombinant AChE. By using 1-naphthol as a marker and 1-naphthol acetate as a substrate, we enhanced sensitivity, selectivity, precision, and time. Our method can provide new opportunities in this industry.

Also, 1-Naphthol acetate is hydrolyzed by both enzymes, such as AChE in RBC/whole blood while BuChE/PChE hydrolyzed in plasma only, but we were focused only on AChE activity on RBC/whole blood samples and not focused on BuChE in plasma. BuChE activities are measured not only to evaluate inhibition by OPs or CMs but also to diagnose a wide range of physiological or pathological conditions reflected in BuChE activities. These measurements are done in serum or plasma and not in whole blood. Therefore, more data on BuChE than AChE activities are available, and more methods have been developed for the analysis of plasma or serum than in whole blood [36]. That's why we used to focus on AChE activity on isolated RBC from whole blood. Therefore, in this manuscript, we measured only the AChE activity on whole blood but did not focus on the BuChE activity in the plasma. We have not measured butyrylcholinesterase activity because we have not used plasma samples, therefore, there we have not considered the activity of butyrylcholine esterase.

Only 20 μL Phosphate buffer was used as a blank which is very less volume there is no question of precipitation in the column or elution. After adding the acetonitrile to the reaction mixture, the phosphate compound present in the buffer precipitated in the reaction mixture after that it was centrifuge and supernatant was collected and filtered with 0.2-micron syringe filter so filtrate is free from cellular debris and other impurities.

This simple finger prick method (since the volume of blood is 10 μL which is similar to the volume obtained from finger prick) would be enough to read the toxicity of an individual and the most important aspect of this method is the time required for getting the results is very short in comparison to any other, which enables us to start treatment of an individual for toxicity depending upon the nature of exposure and magnitude. Till date, this type of facility or

method which checks the toxicity of a patient instantly does not exist in the world. AChE has been primarily studied to estimate the levels of toxicity by OP's and carbamates affecting nerve agents for a long time, while BuChE could also be an interesting parameter to study. However, the present method has been worked on AChE and an in-silico study by Chowdhary S et al., 2018 [34] has reported that the interactions observed in the most favorable Total Interaction Energy pose of Acetylcholine and 1-NA supporting our study that the 1-Naphthol acetate would be a most favorable to Ache over BuChE. This reported method is very effective and easy to run for the analysis of toxicity due to metals and pesticides as the blood samples required is very less, and the remaining sample can be stored and used for other parameters. This method would be enough to read the toxicity of an individual and the most important aspect of this method is the time required for getting the results is very short in comparison to any other, which makes us to start treatment for an individual for toxicity depending upon the nature of exposure, the magnitude, till date this type of facility or method which checks the toxicity of a patient instantly is not existing in world.

## Conclusions

The reported method showed that the 1-naphthol acetate as substrate and 1-naphthol as a product/marker are the best compounds for AChE enzyme assay and quantification of different enzyme related toxicants. Introducing the new biomarker (1-naphthol) in the present method will add to the existing knowledge in the area of enzyme-related toxicant quantification and enzyme estimation due to the high need for a good biomarker and simple method in the field of toxicology. We have developed a prototype approach for estimating AChE in human blood on exposure to O.P., and carbamate exposure using only 10μL of blood sample, and easy extraction procedure made this technique, is quick and robust, and has excellent sensitivity. Furthermore, the approach may be used to measure the AChE in blood of people exposed to Ops, carbamates, and contaminants in water that effect the AChE levels in the human body by monitoring the conversion of the substrate 1-naphthol acetate to 1-naphthol. This reported prototype, a method, would also help to measure exposure and effect which are crucially important to give the prognosis in real- or near real-time to save the life of the exposed person. Furthermore, the reported method is sensitive, accurate, reproducible and easy to run for the analysis of toxicity due to carbamate, metals and pesticides. In addition, only 10μLof blood is the target sample which could provide instant and accurate result. In future, this method could be considered as a milestone and could fill up the gap between analytical toxicology, mechanistic toxicology and clinical toxicology, and would correlate with epidemiological toxicology.

## Acknowledgments

The authors would also like to take this great opportunity to express their heartfelt gratitude to our Secretory, Department of health research, Ministry of Health Government of India for funding the project. The authors would also express their thankfulness to Director, National Institute of Nutrition for the support and motivation to work on the project. The authors are grateful to all participants and their families. The all authors thank to ICMR and NIN for providing the facilities and financial support to carryout work.

## Author Contributions

**Conceptualization:** Sukesh Narayan Sinha.

**Data curation:** Ramakrishna Ungarala, Dileshwar Kumar, Rajendra Sangaraju.

**Formal analysis:** Ramakrishna Ungarala, Dileshwar Kumar.

**Funding acquisition:** Sukesh Narayan Sinha.

**Investigation:** Sukesh Narayan Sinha, Sarita Kumar.

**Methodology:** Sukesh Narayan Sinha, Ramakrishna Ungarala, Rajendra Sangaraju.

**Project administration:** Sukesh Narayan Sinha.

**Resources:** Sukesh Narayan Sinha.

**Supervision:** Sukesh Narayan Sinha, Sarita Kumar.

**Validation:** Sukesh Narayan Sinha, Sarita Kumar.

**Writing – original draft:** Ramakrishna Ungarala, Dileshwar Kumar.

**Writing – review & editing:** Sukesh Narayan Sinha, Ramakrishna Ungarala, Dileshwar Kumar, Rajendra Sangaraju, Sarita Kumar.

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
