## [Decision Letter · Decision Letter 0]

14 Nov 2022

PONE-D-22-30881A novel RP-HPLC method for quantification of cholinesterase activity in human blood: An application for assessing organophosphate and carbamate insecticide exposurePLOS ONE

Dear Dr. Sinha,

Thank you for submitting your manuscript to PLOS ONE. After careful consideration, we feel that it has merit but does not fully meet PLOS ONE’s publication criteria as it currently stands. Therefore, we invite you to submit a revised version of the manuscript that addresses the points raised during the review process.

Please submit your revised manuscript by Dec 29 2022 11:59PM after you have tested plasma in your activity assay. If you will need more time than this to complete your revisions, please reply to this message or contact the journal office at plosone@plos.org. Please include the following items when submitting your revised manuscript:

We look forward to receiving your revised manuscript.

Kind regards,

Oksana Lockridge, Ph.D.

Academic Editor

PLOS ONE

Journal Requirements:

"The study has been funded by the Department of Health Research (F.No. V. 25011/539-HRD/2016-HR, Ministry of Health and Family Welfare, Govt. of India."

Reviewers' comments:

Reviewer's Responses to Questions

**Comments to the Author**

1. Is the manuscript technically sound, and do the data support the conclusions?

Reviewer #1: Partly

2. Has the statistical analysis been performed appropriately and rigorously? 

Reviewer #1: Yes

3. Have the authors made all data underlying the findings in their manuscript fully available?

Reviewer #1: Yes

4. Is the manuscript presented in an intelligible fashion and written in standard English?

Reviewer #1: Yes

5. Review Comments to the Author

Reviewer #1: Summary: An assay for cholinesterase activity in human blood was developed using naphthyl acetate as substrate. Products were detected and quantified by HPLC. Farmers exposed to organophosphorus pesticides had lower activity compared to non-farmers. An individual who ingested an organophosphorus pesticide in an attempt at suicide had very low cholinesterase activity.

Major comment: . A major fallacy in the present work is the notion that hydrolysis of naphthyl acetate is a property of acetylcholinesterase. The published literature reports that butyrylcholinesterase in human plasma hydrolyzes naphthyl acetate, but that acetylcholinesterase in red blood cells does not. There was no attempt to determine whether the activity is in plasma or in red cells. It is acceptable to devise a new method to assay cholinesterase activity. However, the comments in the manuscript about the Ellman assay, indicate the authors have no experience with the Ellman assay. Criticizing the time-honored Ellman assay does not benefit the present report.

1. Introduction: The statement that the Ellman assay does not provide speedy results is incorrect. I use the Ellman assay and get results in 40 seconds.

2. 1-naphthol A should be written 1-naphthol acetate

3. Typing error “Using 1-naphthol acetate as a substrate instead of AChE is expected to be a reliable way to measure cholinesterase activity, avoiding the drawbacks of AChE[34,35]”. In this sentence AChE should be acetylthiocholine.

4. typing error? “4.5 m particle size”. Should this be 4.5 µ particle size?

5. Please clarify the meaning of “blood”. Is it whole blood or plasma or serum? If whole blood, has it been stored frozen? Red blood cells lyse when whole blood is frozen. The lysed red blood cells release hemoglobin. Did hemoglobin precipitate out of solution when you stopped the reaction with acetonitrile?

6. Naphthol acetate is hydrolyzed by butyrylcholinesterase, but not by acetylcholinesterase. P.W. Zapf and C.M. Coghlan (1973) Clinica Chimica Acta 44, 237-242. A kinetic method for the estimation of pseudocholine esterase using naphthyl acetate substrate.

The introduction assumes you measured AChE activity, but this assumption is incorrect.

You measured butyrylcholinesterase activity. In the older literature, butyrylcholinesterase was called pseudocholinesterase. Please modify the text to acknowledge you measured BChE activity.

7. Section 2.6. Please explain what you mean by “These samples were extracted”

8. Section 2.5. Please describe the filter that was used to prepare samples for HPLC. Was it a paper filter? Did the filter separate all the proteins from the substrate and product? Blood contains many small molecules. How did you separate contaminating small molecules from naphthol acetate and naphthol?

9. Table 1 cites reference 54 for hydrolysis of naphthyl acetate by acetylcholinesterase. I checked reference 54 and found that the substrate was acetylthiocholine and the enzyme was a commercial preparation of AChE. Naphthyl acetate was not used in reference 54.

10. “However, this technique has their own limitation, as well as several mL of blood is necessary”. This statement is incorrect. Ellman uses 50 µl AChE in a 3 ml assay. I use 10 µl plasma in a 2 ml Ellman assay.

11. “the possibility of reaction between butyl acetylcholine esterase and 1-napthol acetate is minimal, while similar observation was observed earlier [51-53].” References 51-53 do not support your opinion that AChE rather than BChE is responsible for hydrolysis of naphthyl acetate in your assay. It is suggested that you check the specificity of your assay by separating plasma from red blood cells in a blood sample that has never been frozen, and then measure hydrolysis of naphthyl acetate by plasma using your method. Plasma contains BChE. You will find high activity in plasma.

12. “However, Bche has been studied primarily concerning toxicity caused by drug and drug-drug interactions.” This is an outdated idea. At present BChE is the gold standard for assaying exposure to nerve agents and organophosphorus pesticides.

13. “The steric hinderance due to the butyl structure in the Bche also interferes and prevents the formation on 1-Naphthol from 1 Naphthol Acetate which clearly explains the effect of Bche activity in blood. In case of butyl acetylene cholinesterase, the spatial arrangement of atoms is sterically crowded.” The authors have misunderstood the literature on the crystal structures of AChE and BChE. AChE has a very narrow channel for accommodating substrates and inhibitors. In contrast BChE has a wide open area leading to the active site gorge and a large space at the bottom of the active site gorge. Therefore, BChE can interact with large molecules, but AChE is restricted to interacting with small molecules.

14. Please check the author names in reference 36 and reference 40

15. The name of author Simas in reference 40 is written Simas, B.C.A. but in reference 41 is written Simas, A.B.C.

6. PLOS authors have the option to publish the peer review history of their article (what does this mean?). If published, this will include your full peer review and any attached files.

Reviewer #1: No

---

## [Author Response · Author response to Decision Letter 0]

17 Nov 2022

Major comment: A major fallacy in the present work is the notion that hydrolysis of naphthyl acetate is a property of acetylcholinesterase. The published literature reports that butyrylcholinesterase in human plasma hydrolyzes naphthyl acetate, but that acetylcholinesterase in red blood cells does not. There was no attempt to determine whether the activity is in plasma or in red cells. It is acceptable to devise a new method to assay cholinesterase activity. However, the comments in the manuscript about the Ellman assay, indicate the authors have no experience with the Ellman assay. Criticizing the time-honored Ellman assay does not benefit the present report.

Response: We thank the reviewer for their valuable comments and helping our improve the manuscript and helping us understand the critical points involved for the manuscript. We have performed the assay using water as a sample and then adding AChE to understand the hydrolysis of 1-naphthol acetate. We observed that hydrolysis has been occurred and the conversion to 1-naphtol has been identified on the HPLC chromatogram. The same has been performed using buffer solution with and without AChE and confirmed its conversion by 1-naphthol acetate (Page 13, Section 3.2). Also, an in-silico done by Chowdary S et al. in 2018 has reported that 1-napthol acetate is more favourable to Ache over Bche which was observed in our study as well. We apologise the criticism on the Ellman assay, the method has been a gold standard for many years, our only intention was to report that HPLC method is a more confirmatory method than the calorimetric method (Page 20, Paragraph 1)

Comment 1: Introduction: The statement that the Ellman assay does not provide speedy results is incorrect. I use the Ellman assay and get results in 40 seconds.

Response: We have corrected the statement in the revised method and removed the statement. The Ellman assay is a calorimetric method where as our method is a chromatographic method which is a confirmatory result as compared to calorimetric method. 

Comment 2: 1-naphthol A should be written 1-naphthol acetate

Response: We have abbreviated1-naphthol A to 1-naphthol acetate as per the reviewer’s suggestion.

Comment 3: Typing error “Using 1-naphthol acetate as a substrate instead of AChE is expected to be a reliable way to measure cholinesterase activity, avoiding the drawbacks of AChE [34,35]”. In this sentence AChE should be acetylthiocholine.

Response: We thank the reviewer for the correction, we have corrected the error and included in the revised manuscript as per the suggestion. 

Comment 4: typing error? “4.5 m particle size”. Should this be 4.5 µ particle size?

Response: The typing error has been corrected in the revised manuscript.

Comment 5: Please clarify the meaning of “blood”. Is it whole blood or plasma or serum? If whole blood, has it been stored frozen? Red blood cells lyse when whole blood is frozen. The lysed red blood cells release hemoglobin. Did hemoglobin precipitate out of solution when you stopped the reaction with acetonitrile?

Response: Thank you for comment. The authors would like to convey the response that clarifies the blood, i.e., The blood samples were collected from venous blood into test tubes. Capillaries and test tubes should be heparinized (to prevent blood clotting) and dried (to prevent uncontrolled sample dilution). To prevent contamination of the samples by OPs and carbamates during collection, the skin must be cleaned before sampling. We also focused on AChE activity in whole blood as well as in erythrocytes, we have used whole blood for the analysis. The whole blood has been lysed upon addition of acetonitrile and the RBCs were filtered through 0.2µ syringe filters and the filtrate was used for HPLC analysis

Assay optimization in whole blood

The basic methodology is simple. To check the enzyme activity buffer solutions with 1-naphthol and 1-naphthol acetate were injected separately into the HPLC system and no conversion of substrates was observed. We have also used phosphate buffer with and without ChE to observe the 1-napthol acetate conversion to 1-napthol which clearly indicates that 1-napthol acetate is being converted to 1-napthol by acetylcholinesterase. Further, blood samples from people of our institute who are not exposed to any pesticide sprays were used for observing the reaction and impact of ChE in conversion of the substrate. The peak areas of the substrate during conversion has been represented in the Figure-4. The curves shows the conversion of 1-naphthol acetate to 1-naphthol at different time points. The concentration of 1-naphthol conversion in blood at different time intervals in represented in Figure-5. 

Additionally, to prove our method the reaction was also carried out with RBC with 1-napthol acetate. The RBC were separated from whole blood by centrifuging it at 500 x g for 10 min at 4 degrees C. Aspirate the supernatant (plasma) and add cell wash buffer to the erythrocyte pellet. Discard the supernatant and repeat washing with 2 ml of PBS in order to remove the residual plasma and any residual OP or carbamates that may be present in the blood (Reiner, Elsa, (2006). AChE activity was measured either in unwashed erythrocytes or in erythrocytes that had been washed with buffer or saline in order to remove the residual plasma and any residual inhibitors or oximes that may be present in the blood. In this experiment, we collected erythrocytes (whole blood without plasma) kept at room temperature for use in the experiment. After collecting packed erythrocyte samples and adding 1-naphthyl acetate, the reaction mixture was filtered with 0.22 microns, and the filtrate was collected and loaded into the RP-HPLC. The samples were not frozen or stored at a cold or cool temperature, so there was no risk of hemolyses in the sample, the red blood cells (RBC) was not lysed, and haemoglobin did not precipitate with acetonitrile. When AChE activities are measured in erythrocytes separated from plasma by centrifugation, they are measured either in unwashed erythrocytes or in erythrocytes that have been washed with buffer or saline in order to remove the residual plasma and any residual inhibitors or oximes that may be present in the blood. {Reiner, Elsa (2006). Toxicology of Organophosphate & Carbamate Compounds || Methods for Measuring Cholinesterase Activities in Human Blood., 199–208. doi:10.1016/B978-012088523-7/50015-6}. We describe the blood collection and assay procedures that were incorporated into the revised manuscript. The isolation of RBC is little-bit difficult task not easy so our method focused on whole blood selected as a matrices. 

Assay optimization of RBC AChE activity in human blood

The RBC was separated from whole blood by centrifuging it at 500 x g for 10 min at 4 degrees C. Aspirate the supernatant (plasma) and add cell wash buffer to the erythrocyte pellet. Discard the supernatant and repeat washing with 2 ml of PBS in order to remove the residual plasma and any residual OP or carbamates that may be present in the blood (Reiner, Elsa, (2006). 

Erythrocyte samples and the substrate (1-naphthol acetate) were put in a buffer at room temperature for 20 minutes. The reaction was stopped when acetonitrile was added, and the samples were filtered with 0.22-micron syringe filter paper. The filtrate was then put into the RP-HPLC system. The conversion of 1-naphthol acetate to 1-naphthol, catalysed by the AChE present in the RBC membrane, was followed by the simultaneous disappearance of the former and the appearance of the latter on the chromatograms. Several concentrations and blood volumes were tried, and a 50 µM concentration of 1-naphthol acetate was standardised for the reaction and estimation. Finally, 10 µL of blood was standardised for the estimation. Subsequently, 10 µL of 50 µM 1-naphthol acetate was added to the blood, followed by 280 µL of phosphate buffer. The reaction was monitored every 5 minutes. The complete conversion of 1-naphthol acetate to 1-naphthol was achieved using a 50 µM concentration in water within 20 minutes of the reaction initiation, which proves that the enzyme is very reactive and responsible for the conversion of 1-naphthol acetate to 1-naphthol. Similarly, in 20 µL of phosphate buffer was run on RP-HPLC, no peak was observed, which acts as a control. 

Comment 6: Naphthol acetate is hydrolyzed by butyrylcholinesterase, but not by acetylcholinesterase. P.W. Zapf and C.M. Coghlan (1973) ClinicaChimica Acta 44, 237-242. A kinetic method for the estimation of pseudocholine esterase using naphthyl acetate substrate.

The introduction assumes you measured AChE activity, but this assumption is incorrect.

You measured butyrylcholinesterase activity. In the older literature, butyrylcholinesterase was called pseudocholinesterase. Please modify the text to acknowledge you measured BChE activity.

Response: In 50-µl clean water (LC-MS Grade), a solution of 50-µmole of 1-napthol acetate (100-µl) was added and followed by AChE (2 unit/µl) in phosphate buffer (pH-6.88) (400µl). After different time interval the reaction was stopped using 500-µl Acetonitrile and filtered using a 0.2µ syringe filter and a 10µl reaction mixture was injected in HPLC and confirmed that AChE was involved in the conversion. 

Also 1-NA can be an attractive chromogenic substrate than the Acetylthiocholine or Butyryl thiocholine for the measurement of AChE activity. 1-NA to be a better alternative substrate for AChE activity than acetylthiocholine (ATCh) in terms of lower Km value. Its specificity appeared at least similar to ATCh. Therefore, the authors proposed that 1-NA can be an attractive chromogenic substrate for the measurement of AChE activity, and it possesses the potential to detect organophosphorus pesticide (OP) poisoning. The same has been observed in the study by Chowdary S et al., 2018

Also, 1-Naphthol acetate is hydrolyzed by both enzymes, such as AChE in RBC/whole blood while BuChE/PChE hydrolysed in plasma only (Reiner, Elsa (2006). Toxicology of Organophosphate & Carbamate Compounds || Methods for Measuring Cholinesterase Activities in Human Blood.), but we were focused only on AchE activity on RBC/whole blood samples and not focused on BuChE in plasma.

BuChE activities are measured not only to evaluate inhibition by OPs or CMs but also to diagnose a wide range of physiological or pathological conditions reflected in BuChE activities. These measurements are done in serum or plasma and not in whole blood. Therefore, more data on BuChE than AChE activities are available, and more methods have been developed for the analysis of plasma or serum than in whole blood (Reiner, Elsa (2006). Toxicology of Organophosphate & Carbamate Compounds || Methods for Measuring Cholinesterase Activities in Human Blood.). That’s why we used to focus on AChE activity on isolated RBC from whole blood. Therefore, in this manuscript, we measured only the AChE activity on whole blood but did not focus on the BuChE activity in the plasma; this paper assumes we only focused on AchE activity. We have not measured butyrylcholinesterase activity because we have not used plasma samples, therefore, there we have not considered the activity of butyrylcholine esterase (Page 19).

Comment 7: Section 2.6. Please explain what you mean by “These samples were extracted”

Response: All the blood samples collected from Healthy people, Farmers and suicidal patients were used for extraction and analysis on the developed method.

Comment 8: Section 2.5. Please describe the filter that was used to prepare samples for HPLC. Was it a paper filter? Did the filter separate all the proteins from the substrate and product? Blood contains many small molecules. How did you separate contaminating small molecules from naphthol acetate and naphthol?

Response:0.2µ syringe filters were used for filtration. The filtration was carried out by gently applying pressure on the piston of the syringe and the filtrate was collected into 200 µL inserts placed into the 1.5ml Screwcap vials.

Comment 9: Table 1 cites reference 54 for hydrolysis of naphthyl acetate by acetylcholinesterase. I checked reference 54 and found that the substrate was acetylthiocholine and the enzyme was a commercial preparation of AChE. Naphthyl acetate was not used in reference 54.

Response: We tried to emphasise on different methods in Table 1, that has been developed previously for the identification and estimation of acetylcholinesterase. The intention of our inclusion with reference 54 was about the method that the authors employed for identifying acetylcholinesterase on HPLC where they used Methanol: Water: Diethyl Amine in the ration of 40:60:0.05 while our method was employed to identify 1-napthy acetate with simpler mobile phase of acetonitrile–water in the ratio of 45:55 at 1ml/min flow rate which is quite solvent saving and easier to employ.

Additionally, we thank you for your valuable suggestion, We would to explain that, as done by Chowdhary S, Bhattacharyya R, Banerjee D, Biochime 2018, they observed Erythrocyte acetylcholinesterase (AChE) is a preferred biomarker for the detection of organophosphorus poisoning.The in silico approach was performed to understand the fitness and the Total Interaction Energy (TIE) of substrates for AChE. The alternative substrates for AChE were screened in terms of high Goldscore and favorable TIE in comparison to acetylcholine (ACh)-AChE complex and other relevant esterases. Among the screened substrates, 1-Naphthyl acetate (1-NA) exhibited the most favorable interaction with AChE in terms of highest TIE and corresponding high Goldscore. The Molecular Dynamic (MD) simulation of the 1-NA-AChE complex showed a stable complex formation over a period of 5 ns. The results obtained in the in silico studies were validated in vitro using pure erythrocyte AChE and hemolysate.

 1-NA to be a better alternative substrate for AChE activity than acetylthiocholine (ATCh) in terms of lower Km value. Its specificity appeared at least similar to ATCh. Therefore, the authors proposed that 1-NA can be an attractive chromogenic substrate for the measurement of AChE activity, and it possesses the potential to detect organophosphorus pesticide (OP) poisoning}”.

Comment 10: “However, this technique has their own limitation, as well as several mL of blood is necessary”. This statement is incorrect. Ellman uses 50 µl AChE in a 3 ml assay. I use 10 µl plasma in a 2 ml Ellman assay.

Response: We agree with the reviewer comment and apologise for our statement, we have deleted the statement in the revised method. However, these days the most applied technique to determine cholinesterase activity is Ellman’s colorimetry. It is suitable for the assessment of biological samples, for the identification of AChE and the determination of their efficacy. But its limitation lies in strong interaction with hemoglobin during whole blood sample analysis. Furthermore, Ellman’s colorimetry cannot be uncritically used to assay the oxime reactivators. It may not be used in various protocols for the determination of pesticide or nerve agent exposure. This method used only for hemoglobin-rich samples. Johnson and Russel reported radiometric assay is a practical approach for protocols where high sensitivity is needed. Fluorometric methods are useful for sensitive determination within kinetic studies at very low enzyme concentrations. Similarly, Biosensors seem to be suitable for OP and carbamate identification. The AChE activity determination approach needs to fulfill several requirements like sufficiently sensitive, selectivity for AChE activity is necessary. Our method is capable for the assessment of biological samples as well as for kinetic studies and new drug discovery at very low level. Earliest techniques were developed mainly for overall AChE activity determination without accuracy. All techniques used either natural substrate AChE or artificial substrates. Our new reported method extensively improved in respect of sensitivity, selectivity, great precision, and time factor by introducing new biomarker 1-Napthol and 1-napthol acetate as a substrate. Great potential lies in use of our methods, which creates new possibilities in this field.

This method developed on the basis for a variety of tests that might have used to measure biological effect. The principles that developed method here could be used in clinical settings as enzyme activities that are impaired by specific exposures are discovered. Arsenic, for example, is known to interact with a wide variety of sulfur containing proteins. This work will also have an impact on exposure to polycyclic aromatic hydrocarbons (Page 17). 

Comment 11: “the possibility of reaction between butyl acetylcholine esterase and 1-napthol acetate is minimal, while similar observation was observed earlier [51-53].” References 51-53 do not support your opinion that AChE rather than BChE is responsible for hydrolysis of naphthyl acetate in your assay. It is suggested that you check the specificity of your assay by separating plasma from red blood cells in a blood sample that has never been frozen, and then measure hydrolysis of naphthyl acetate by plasma using your method. Plasma contains BChE. You will find high activity in plasma.

Response: We agree with the reviewer comments that plasma has high activity from BChE, however AChE has been more favourable in interaction towards 1-naphthol acetate as compared to BChE. The same has been studied by Chowdary S et al., 2018 and the in-silico interaction with different esterase have revealed the same as reported in our study. We have also modified the references to support our statement and removed other references.

Comment 12: “However, Bche has been studied primarily concerning toxicity caused by drug and drug-drug interactions.” This is an outdated idea. At present BChE is the gold standard for assaying exposure to nerve agents and organophosphorus pesticides.

Response: The statement could be true that Bche was primarily studied for toxicity caused by BChE may be an old statement, however our method has been worked on AchE and an in-silico study by Chowdary S et al., 2018 has reported that the interactions observed in the most favorable Total Interaction Energy pose of ACh and 1-NA supporting our study that the 1-Naphthol acetate would be a most favourable to Ache over Bche. As per the comment from the reviewer we have deleted the previous statement modified the statement with the above response.

Comment 13: “The steric hinderance due to the butyl structure in the Bche also interferes and prevents the formation on 1-Naphthol from 1 Naphthol Acetate which clearly explains the effect of Bche activity in blood. In case of butyl acetylene cholinesterase, the spatial arrangement of atoms is sterically crowded.” The authors have misunderstood the literature on the crystal structures of AChE and BChE. AChE has a very narrow channel for accommodating substrates and inhibitors. In contrast BChE has a wide-open area leading to the active site gorge and a large space at the bottom of the active site gorge. Therefore, BChE can interact with large molecules, but AChE is restricted to interacting with small molecules.

Response: We apologise for our comment made previously and agree with the reviewer comment on the active sites, we have deleted the statement on steric hinderance effect on the substrates. We are thankful to the reviewer for the valuable comment and understanding of the interaction pockets for us.

Comment 14: Please check the author names in reference 36 and reference 40

Response: We thank the reviewer for the comment, we have corrected the errors thet were found in the references.

Comment 15: The name of author Simas in reference 40 is written Simas, B.C.A. but in reference 41 is written Simas, A.B.C.

Response: We have cross checked with the citations again and the same has been corrected in the revised manuscript. The author in reference 40 was cited as Simas, B.C.A while in reference 41 it was represented as Simas, A.B.C. as extracted from the published citations.

---

## [Decision Letter · Decision Letter 1]

23 Nov 2022

PONE-D-22-30881R1A novel RP-HPLC method for quantification of cholinesterase activity in human blood: An application for assessing organophosphate and carbamate insecticide exposurePLOS ONE

Dear Dr. Sinha,

Thank you for submitting your manuscript to PLOS ONE. After careful consideration, we feel that it has merit but does not fully meet PLOS ONE’s publication criteria as it currently stands. Therefore, we invite you to submit a revised version of the manuscript that addresses the points raised during the review process.

The text contains many words that run together without a space between words.  Please correct these typing errors.  Also, please check your manuscript for grammar and spelling errors.  Language errors will delay acceptance of your manuscript.  

Please include the following items when submitting your revised manuscript:A rebuttal letter that responds to each point raised by the academic editor and reviewer(s). You should upload this letter as a separate file labeled 'Response to Reviewers'.A marked-up copy of your manuscript that highlights changes made to the original version. You should upload this as a separate file labeled 'Revised Manuscript with Track Changes'.An unmarked version of your revised paper without tracked changes. You should upload this as a separate file labeled 'Manuscript'.

We look forward to receiving your revised manuscript.

Kind regards,

Oksana Lockridge, Ph.D.

Academic Editor

PLOS ONE

Reviewers' comments:

Reviewer's Responses to Questions

**Comments to the Author**

1. If the authors have adequately addressed your comments raised in a previous round of review and you feel that this manuscript is now acceptable for publication, you may indicate that here to bypass the “Comments to the Author” section, enter your conflict of interest statement in the “Confidential to Editor” section, and submit your "Accept" recommendation.

Reviewer #1: (No Response)

2. Is the manuscript technically sound, and do the data support the conclusions?

Reviewer #1: (No Response)

3. Has the statistical analysis been performed appropriately and rigorously? 

Reviewer #1: (No Response)

4. Have the authors made all data underlying the findings in their manuscript fully available?

Reviewer #1: (No Response)

5. Is the manuscript presented in an intelligible fashion and written in standard English?

Reviewer #1: No

6. Review Comments to the Author

Reviewer #1: 1. The name Chowdhary is misspelled on pages 68, 81, 85, and 88.

2. Page 51 PDF-based detection should be PDA-based detection

3. Page 53 section 2.3.1. Isocratic elution was with water-acetonitrile 55:45. However, the last sentence in section 2.3.1 states “All measurements were carried out at room temperature in phosphate buffer (pH-6.88) and acetonitrile”. Please clarify this inconsistency. Did you elute with water-acetonitrile or buffer-acetonitrile? If the eluant contained buffer, please indicate the concentration of phosphate buffer. This is a concern because phosphate buffer may precipitate out of solution in the presence of acetonitrile.

4. Page 55 section 2.5. “a solution of 50-µmole of 1-napthol acetate (100-µl)” I think you mean 100 µl of a 50 µM solution of 1-naphthyl acetate? 50 µmole in 100 µl calculates to 0.5 moles/liter, a concentration much too high for your assay.

5. Page 55 section 2.5. “AChE (2 unit/µl)” implies that you used a commercial preparation of AChE whose activity was defined by the supplier. If this guess is correct, please add the name of the supplier, the catalog number and the AChE species (human, horse, electric eel?)

6. Before you describe assays that use blood, you need to describe what you mean by blood. It is important for the reader to know whether you used whole blood (never frozen) or whole blood that was frozen and thawed, or isolated red blood cells. A separate section describing blood should be included before section 2.3.

7. Section 2.5 is entitled “Assay optimization in water”. This title is confusing because the assay contains 400 µl phosphate buffer plus 100 µl of naphthyl acetate in acetonitrile plus 50 µl water. Please clarify why 50 µl water optimizes the assay.

8. Page 56 and page 57 “Finally, 10 µL of blood sample was standardized for the estimation. Subsequently, 10 µL of 50 µM 1-naphthol acetate was added to the blood followed by 280 µL of phosphate buffer.” This order of addition is a problem. If you added 10 µl of 50 µM naphthyl acetate dissolved in acetonitrile to 10 µl blood, the acetonitrile would inactivate the activity of AChE. A more likely order of addition is 10 µl blood added to 280 µl buffer, followed by 10 µl of 50 µM 1-naphthyl acetate.

9. Page 56. “The complete conversion of 1-naphthol acetate to 1-naphthol was achieved using 50 µmole concentration in water”. According to page 52 section 2.2, 50 µM 1-naphthyl acetate was in acetonitrile, but on page 56 you say the naphthyl acetate was in water. Was your naphthyl acetate solution in acetonitrile or in water? Is naphthyl acetate soluble in water at 50 µM?

10. Page 56. “Similarly, in 1 mL phosphate buffer run on HPLC.” I do not understand this sentence.

11. Page 56. The description of the method for preparing red blood cells should be in a separate section, before section 2.3.

12. Page 57 “All the blood samples collected from healthy volunteers, farmers and intentionally ingested people were extracted”. Please explain what you mean when you say blood samples were extracted. The word extraction means you removed something from the blood. My understanding of the text is that you used whole blood diluted into buffer. I do not see an extraction step.

13. Table 1 title. Suggested title: Literature review of HPLC methods for assaying AChE activity and inhibition.

14. Sections 3.1.3, 3.1.4, 3.1.5, 3.1.6 and 3.1.7. Were these values measured in the presence of blood or in the absence of blood? Please clarify this point. I am guessing that no blood or AChE was present in the results for sections 3.1.3 to 3.1.7. Please add a statement that the assays did not include AChE or blood.

15. Figure 4 and 5 legends. I think Figures 4 and 5 represent AChE activity, but I am not certain. If they do represent AChE activity, please modify the figure legends as follows: “1-naphthol acetate conversion to 1-naphthol by AChE in blood at different time intervals.”

16. Figure 5 legend “Concentration of 1-naphthol formed by AChE in blood at different time intervals.”

17. Tables 4 and 5. Please define units/ml. Is it µmoles naphthyl acetate hydrolyzed per ml?

18. Table 4 and 5. Which data in which figures were used to calculate units of activity?

19. Please use the Spell Check feature in Microsoft Word to correct the many errors in the text.

7. PLOS authors have the option to publish the peer review history of their article (what does this mean?). If published, this will include your full peer review and any attached files.

Reviewer #1: No

---

## [Author Response · Author response to Decision Letter 1]

26 Nov 2022

Responses to Reviewers

Editor and Reviewer Comments:

According to the Editor’s suggestion, the author followed the PLOS ONE journal authors’ guidelines and revised the whole manuscript as per guidelines.

The reviewer’s comments are in blue color and the author’s responses are in black color. The author incorporated some changes in the revised manuscript and highlighted them in yellow color.

Response to Reviewer #1

Thank you very much for considering this manuscript. The author is grateful for the suggestions and recommendations made by you to improve the quality of our manuscript. The manuscript is now revised carefully according to the Editor and reviewer’s suggestions and included it in the detailed point-to-point answers mentioned below. 

Comment 1: The name Chowdhary is misspelled on pages 68, 81, 85, and 88.

Response to Reviewer #1 comment 1: We thank the reviewer for the correction, we have corrected the error in citation representation and included it in the revised manuscript as per the suggestion. According to your suggestion, We corrected the spelling of Chowdary in Line no. 385, page No. 19 ; Line no. 442, page no. 22, and citation no. 34. According to your suggestion

Comment 2: Page 51 PDF-based detection should be PDA-based detection

Response to Reviewer #1 comment 2: We thank the reviewer for the correction, we have corrected the error and included it in the revised manuscript as per the suggestion. We have corrected the error Line No. 79, page No. 4 in the revised manuscript.

Comment 3: Page 53 section 2.3.1. Isocratic elution was with water-acetonitrile 55:45. However, the last sentence in section 2.3.1 states “All measurements were carried out at room temperature in phosphate buffer (pH-6.88) and acetonitrile”. Please clarify this inconsistency. Did you elute with water-acetonitrile or buffer-acetonitrile? If the eluant contained buffer, please indicate the concentration of phosphate buffer. This is a concern because phosphate buffer may precipitate out of solution in the presence of acetonitrile.

Response to Reviewer #1 comment 3: We apologize for the inconsistency in the data representation; we have corrected and rewritten the Section “liquid chromatography conditions” to eliminate the errors, and it has been included in the revised manuscript. All substances were separated chromatographically using a liquid chromatography with a C18 reversed-phase column (150 x 4.6 mm ID, 4.5 µ particle size). The Shimadzu auto-sampler was used to inject samples (20 µL) into the device. Then a 20µL reaction mixture was injected into the RP-HPLC. All measurements were carried out at room temperature. The isocratic composition of mobile phase water–acetonitrile (55: 45, v/v) pumped at a flow rate of 1 mL/min provided the best sensitivity and separation of naphthol compounds. All the standards of 1-naphthyl acetate were prepared in acetonitrile. The temperature in the column oven was kept constant at 25 °C. At a wavelength of 280 nm, the absorbance was measured. and the data were integrated into the software. Generally, the enzyme reaction takes place at pH 7 in the buffer solution. Therefore, the phosphate buffer was used to carry out the reaction in blood samples, and acetonitrile was used to stop the reaction to optimize the kinetics of the reaction at different time periods. Phosphate buffer (pH 6.88) medium was used to carry out the reaction in blood samples and acetonitrile was used for stopped the reaction in blood samples. Only 20 µL Phosphate buffer was used which is very less volume there is no question of precipitation in the column or elution and after adding the acetonitrile to the reaction mixture the phosphate compound present in the buffer precipitated in the reaction mixture after that it was centrifuge and supernatant was collect and filtered with 0.2-micron syringe filter so filtrate is free from cellular debris. As per your valuable suggestion, we incorporated it in the revised manuscript with lines no: 130-142, page no:6-7.

Comment 4: Page 55 section 2.5. “a solution of 50-µmole of 1-napthol acetate (100-µL)” I think you mean 100 µL of a 50 µM solution of 1-naphthyl acetate? 50 µmole in 100 µL calculates to 0.5 moles/liter, a concentration much too high for your assay.

Response to Reviewer #1 comment 4: We thank the reviewer for the correction; we have corrected the error in the revised manuscript. To confirm that AChE is involved in the conversion of 1-naphthol acetate to 1-naphthol, the following assay was performed in 50 µL clean water (LC-MS grade) add 50 µL solution of 50 µM of 1-naphthol acetate was added and followed by AChE (2 unit/µL) in 400µL phosphate buffer (pH 6.88). After different time intervals, the reaction was stopped using with 500 µL acetonitrile and filtered using a 0.2µ syringe filter and a 20µL reaction mixture was injected into RP-HPLC. So, the concentration of 1-naphthol acetate was not too high. As per your valuable suggestion, we incorporated it in the revised manuscript with lines no: 187-192, page no:9.

Comment 5: Page 55 section 2.5. “AChE (2 unit/µL)” implies that you used a commercial preparation of AChE whose activity was defined by the supplier. If this guess is correct, please add the name of the supplier, the catalogue number and the AChE species (human, horse, electric eel?)

Response to Reviewer #1 comment 5: AChE (Human) was procured from Sigma-Aldrich with CAS-No: 9000-81-1 and product number of C0663. The same was used for the reaction in our manuscript. As per your valuable suggestion, we incorporated it in the revised manuscript with lines no: 105-106, page no:5.

Comment 6: Before you describe assays that use blood, you need to describe what you mean by blood. It is important for the reader to know whether you used whole blood (never frozen) or whole blood that was frozen and thawed, or isolated red blood cells. A separate section describing blood should be included before section 2.3.

Response to Reviewer #1 comment 6: The 2 ml whole blood was collected from healthy volunteers and then subjected to a reaction without freezing the samples. A sample volume of 10 µL from the 2 ml collected blood was used for the reaction. The blood samples were collected from venous blood and placed in test tubes. Capillaries and test tubes should be heparinized (to prevent blood clotting) and dried (to prevent uncontrolled sample dilution). To prevent contamination of the samples by OPs and carbamates during collection, the skin must be cleaned before sampling. Similarly, The RBC were separated from whole blood by centrifuging it at 500 x g for 10 min at 4 degrees C. Aspirate the supernatant (plasma) and add cell wash buffer to the erythrocyte pellet. Discard the supernatant and repeat washing with 2 ml of PBS (Phosphate buffered saline) in order to remove the residual plasma and any residual OPs or carbamates that may be present in the blood [36]. As per your valuable suggestion, we incorporated it in the revised manuscript with lines no: 114-124, page no:6.

Comment 7: Section 2.5 is entitled “Assay optimization in water”. This title is confusing because the assay contains 400 µL phosphate buffer plus 100 µL of naphthyl acetate in acetonitrile plus 50 µL water. Please clarify why 50 µL water optimizes the assay.

Response to Reviewer #1 comment 7: To confirm that AChE is involved in the conversion of 1-naphthol acetate to 1-naphthol, the following assay was performed. 50 µL clean water (LC-MS grade) add with 50 µL solution of 50 µM of 1-naphthol acetate (100 µL) was added and followed by AChE (2 unit/µL) in 400µL phosphate buffer (pH 6.88). After different time intervals, the reaction was stopped using with 500 µL Acetonitrile and filtered using a 0.2µ syringe filter and collect the filtrate of 20µL reaction mixture was injected in RP-HPLC under the same conditions mentioned above. As per your valuable suggestion, we incorporated it in the revised manuscript with lines no: 186-192, page no:9

Comment 8: Page 56 and page 57 “Finally, 10 µL of blood sample was standardized for the estimation. Subsequently, 10 µL of 50 µM 1-naphthol acetate was added to the blood followed by 280 µL of phosphate buffer.” This order of addition is a problem. If you added 10 µL of 50 µM naphthyl acetate dissolved in acetonitrile to 10 µL blood, the acetonitrile would inactivate the activity of AChE. A more likely order of addition is 10 µL blood added to 280 µL buffer, followed by 10 µL of 50 µM 1-naphthyl acetate.

Response to Reviewer #1 comment 8: We thank the reviewer for the comment, we have corrected the error and also included a schematic figure representing the process in the revised manuscript. Firstly 10µL blood was added in 280 µL phosphate buffer was taken and, thereafter 10µL of 50µM 1- naphthol acetate was added and incubated up to 20 minutes, next add 700 µL acetonitrile for stopped the reaction and filtered. 20 µL filtrate was injected to RP-HPLC system. As per your valuable suggestion, we incorporated it in the revised manuscript with lines no: 193-210, page no:10

 This procedure is depicted in the schematic figure below.

Comment 9: Page 56. “The complete conversion of 1-naphthol acetate to 1-naphthol was achieved using 50 µmole concentration in water”. According to page 52 section 2.2, 50 µM 1-naphthyl acetate was in acetonitrile, but on page 56 you say the naphthyl acetate was in water. Was your naphthyl acetate solution in acetonitrile or in water? Is naphthyl acetate soluble in water at 50 µM?

Response to Reviewer #1 comment 9:1-naphthol acetate solution was prepared in acetonitrile, not in water. We apologize for the error in the representation; we have used acetonitrile as the solvent, and the same has been corrected in the revised manuscript. As per your valuable suggestion, we incorporated it in the revised manuscript with lines no:135- 136, pages no:6 -7. 

Comment 10: Page 56. “Similarly, in 1 mL phosphate buffer run on HPLC.” I do not understand this sentence.

Response to Reviewer #1 comment 10: Similarly, in 20 µL of phosphate buffer was injected into the system and run-on RP-HPLC, where no peak was observed, which acts as a control. The same has been corrected in the revised manuscript. 1ml phosphate buffer was not run in RP-HPLC. As per your valuable suggestion, we incorporated it in the revised manuscript with lines no:207-210, page number 10;

Comment 11: Page 56. The description of the method for preparing red blood cells should be in a separate section, before section 2.3.

Response to Reviewer #1 comment 11: We have separated the red blood cells preparation part as per the suggestions of the reviewer. The separation method of RBC is mentioned in 2.2.1 section “Collection of blood and isolation of RBC” 

The 2 ml whole blood was collected from healthy volunteers and then subjected to a reaction without freezing the samples. A sample volume of 10 µL from the 2 ml collected blood was used for the reaction. The blood samples were collected from venous blood and placed in test tubes. Capillaries and test tubes should be heparinized (to prevent blood clotting) and dried (to prevent uncontrolled sample dilution). To prevent contamination of the samples by OPs and carbamates during collection, the skin must be cleaned before sampling. Similarly, The RBC were separated from whole blood by centrifuging it at 500 x g for 10 min at 4 degrees C. Aspirate the supernatant (plasma) and add cell wash buffer to the erythrocyte pellet. Discard the supernatant and repeat washing with 2 ml of PBS (Phosphate buffered saline) in order to remove the residual plasma and any residual OPs or carbamates that may be present in the blood. As per your valuable suggestion, we incorporated it in the revised manuscript with lines no: 114-124, page no:6.

Comment 12: Page 57 “All the blood samples collected from healthy volunteers, farmers and intentionally ingested people were extracted”. Please explain what you mean when you say blood samples were extracted. The word extraction means you removed something from the blood. My understanding of the text is that you used whole blood diluted into buffer. I do not see an extraction step.

Response to Reviewer #1 comment 12: We have corrected the phrase with subjected to “extraction” in the revised manuscript as per the comment from the reviewer. All the blood samples were collected from healthy volunteers, and farmers exposed to the pesticide while spraying, and suicidal patients who ingested pesticides while hospitalized. All blood samples were subjected to the reaction procedure and analysed on the developed method and AChE levels were estimated and mentioned in Fig 1. The blood samples were subjected to reaction and then the conversion was monitored on the HPLC system which we have represented as an extraction. After completion of the reaction for 20 minutes, added with acetonitrile was used to stop the reaction to inhibit AChE enzyme activity on RBC, and thereafter the reaction mixture was filtered using 0.2µ syringe filters for filtration and collect the filtrate and discard the other cellular debris. The 20µL filtrate was injected into the RP HPLC system, which we may have represented as an extraction, where actually it is not an extraction. It is only filtration to remove blood cell-related impurities. We incorporated it in the revised manuscript lines no: 221-232 page no: 11 

Comment 13: Table 1 title. Suggested title: Literature review of HPLC methods for assaying AChE activity and inhibition.

Response to Reviewer #1 comment 13: We have modified the title as per the suggestion of the reviewer and included in the revised manuscript. As per your valuable suggestion, we incorporated it in the revised manuscript with lines no: 241, page no:11.

Comment 14: Sections 3.1.3, 3.1.4, 3.1.5, 3.1.6 and 3.1.7. Were these values measured in the presence of blood or in the absence of blood? Please clarify this point. I am guessing that no blood or AChE was present in the results for sections 3.1.3 to 3.1.7. Please add a statement that the assays did not include AChE or blood.

Response to Reviewer #1 comment 14: We thank the reviewer for the comment. During the method validations AChE or blood were not used. and was incorporated into sections 3.1.3 to 3.1.7 as per reviewer’s suggestions. As per your valuable suggestion, we incorporated it in the revised manuscript with lines no: 145 page no: 7

Comment 15: Figure 4 and 5 legends. I think Figures 4 and 5 represent AChE activity, but I am not certain. If they do represent AChE activity, please modify the figure legends as follows: “1-naphthol acetate conversion to 1-naphthol by AChE in blood at different time intervals.”

Response to Reviewer #1 comment 15: As per the reviewer’s suggestion we have modified the figure legends in the revised manuscript. figure 4 becomes figure 5; we incorporated it in the revised manuscript with lines no: 311 page no:16

• Fig 5. 1-naphthol acetate conversion to 1-naphthol by AChE in blood at different time intervals

Comment 16: Figure 5 legend “Concentration of 1-naphthol formed by AChE in blood at different time intervals.”

Response to Reviewer #1 comment 16: As per the reviewer suggestion we have modified the figure legends in the revised manuscript. figure 5 become figure 6; we incorporated it in the revised manuscript with lines no: 312 page no:16

• Fig 6. Concentration of 1-naphthol formed by AChE in blood at different time intervals

Comment 17: Tables 4 and 5. Please define units/ml. Is it µmoles naphthyl acetate hydrolysed per ml?

Response to Reviewer #1 comment 17: The 1-naphthol acetate was hydrolysed per ml but the enzyme activity was expressed in units/ml. We cannot express the enzyme activity in only ml. In Tables 4 and 5, the enzyme activity was expressed in units/ml. Therefore, it was mentioned.

Comment 18: Tables 4 and 5. Which data in which figures were used to calculate units of activity?

Response to Reviewer #1 comment 18: Table 4 and Table 5 represents the levels and concentrations of AChE in the blood collected from control, Exposed and Intendedly ingested persons which we have not represented in any of the figures in our manuscript. Each batch of analysis was included with a standard 50µM 1-naphthol and 1 naphthol acetate in the analysis. The calculation was represented with the standard concentrations of the particular batch. The concentration of formation of a product that is 1-naphthol was considered for the calculation of enzyme activities which is represented in figure 5.

Comment 19: Please use the Spell Check feature in Microsoft Word to correct the many errors in the text.

Response to Reviewer #1 comment 19: We thank the reviewer for the suggestion and as per the recommendation we have corrected the errors in the text of the revised manuscript by using Spell Check feature in Microsoft Word.

---

## [Decision Letter · Decision Letter 2]

1 Dec 2022

PONE-D-22-30881R2A novel RP-HPLC method for quantification of cholinesterase activity in human blood: An application for assessing organophosphate and carbamate insecticide exposurePLOS ONE

Dear Dr. Sinha,

Thank you for submitting your revised manuscript to PLOS ONE. After careful consideration, we feel that additional changes are required. Therefore, we invite you to submit a revised version of the manuscript that addresses the points raised during the review process. The Discussion section is too long and repetitive.  You can avoid repeating yourself by making a heading for each point you want to make and focusing the discussion on that particular point.  

 Please include the following items when submitting your revised manuscript:A rebuttal letter that responds to each point raised by the academic editor and reviewer(s). You should upload this letter as a separate file labeled 'Response to Reviewers'.A marked-up copy of your manuscript that highlights changes made to the original version. You should upload this as a separate file labeled 'Revised Manuscript with Track Changes'.An unmarked version of your revised paper without tracked changes. You should upload this as a separate file labeled 'Manuscript'.If applicable, we recommend that you deposit your laboratory protocols in protocols.io to enhance the reproducibility of your results. Protocols.io assigns your protocol its own identifier (DOI) so that it can be cited independently in the future. For instructions see: https://journals.plos.org/plosone/s/submission-guidelines#loc-laboratory-protocols. Additionally, PLOS ONE offers an option for publishing peer-reviewed Lab Protocol articles, which describe protocols hosted on protocols.io. Read more information on sharing protocols at https://plos.org/protocols?utm_medium=editorial-email&utm_source=authorletters&utm_campaign=protocols.

We look forward to receiving your revised manuscript.

Kind regards,

Oksana Lockridge, Ph.D.

Academic Editor

PLOS ONE

Journal Requirements:

Reviewers' comments:

Reviewer's Responses to Questions

**Comments to the Author**

1. If the authors have adequately addressed your comments raised in a previous round of review and you feel that this manuscript is now acceptable for publication, you may indicate that here to bypass the “Comments to the Author” section, enter your conflict of interest statement in the “Confidential to Editor” section, and submit your "Accept" recommendation.

Reviewer #1: (No Response)

2. Is the manuscript technically sound, and do the data support the conclusions?

Reviewer #1: (No Response)

3. Has the statistical analysis been performed appropriately and rigorously? 

Reviewer #1: (No Response)

4. Have the authors made all data underlying the findings in their manuscript fully available?

Reviewer #1: (No Response)

5. Is the manuscript presented in an intelligible fashion and written in standard English?

Reviewer #1: (No Response)

6. Review Comments to the Author

Reviewer #1: 1. Page 5 line 109. Please change 1 Mole to 1 M.

2. Page 9 line 190 states that the assay was stopped by addition of 500 µl acetonitrile. However, line 204 and the scheme show the assay was stopped with 700 µl acetonitrile. Please clarify this inconsistency.

3. Page 9 lines 187-192. I do not understand why you think addition of 50 µl water to 50 µl naphthyl acetate and 400 µl AChE in phosphate buffer is a test of the role of AChE in ester hydrolysis. This section would make sense if the assay included no AChE, so that you would be testing the spontaneous hydrolysis of the ester.

4. Page 9 line 186. If you are testing spontaneous hydrolysis of naphthyl acetate, then the title of this section should be changed to “Auto hydrolysis of napththyl acetate in the absence of AChE”

5. Page 10 line 212. Please delete “in 2.6.1”

6. Page 11 line 230. Please define a unit of activity.

7. Page 14 line 351. The statement is incorrect that the Ellman method utilizes 10 µl of plasma sample. The original Ellman method used whole blood. The Ellman method has been adapted to assay BChE activity in plasma samples, but plasma activity was not measured by Ellman in their classic 1961 paper.

8. Page 20 line 396. The statement that your HPLC assay is the best analytical approach is not convincing. You have no experience with other assays, but rely on literature reports to make your claim.

9. Page 20 line 408. “synthetic AChE” do you mean recombinant AChE?

10. The Discussion should be shortened.

7. PLOS authors have the option to publish the peer review history of their article (what does this mean?). If published, this will include your full peer review and any attached files.

Reviewer #1: No

---

## [Author Response · Author response to Decision Letter 2]

2 Dec 2022

Response to Reviewer

Reviewer #1: 

The reviewer’s comments are in blue color and the author’s responses are in black color. The author incorporated some changes in the revised manuscript and highlighted them in red color with track changes.

Thank you very much for your valuable suggestions, and comments, and for considering this manuscript. The author is grateful for the suggestions and recommendations made by you to improve the quality of our manuscript. The manuscript is now revised carefully according to the Editor’s and best reviewer’s suggestions and included it in the detailed point-to-point answers mentioned below. 

Comment 1. Page 5 line 109. Please change 1 Mole to 1 M.

Response to Reviewer #1 comment 1: We thank the reviewer for the correction. According to your suggestion, We corrected the spelling. and we have corrected the error and included it in the revised manuscript (line number 110, page no:6).

Comment 2. Page 9 line 190 states that the assay was stopped by addition of 500 µl acetonitrile. However, line 204 and the scheme show the assay was stopped with 700 µl acetonitrile. Please clarify this inconsistency.

Response to Reviewer #1 comment 2: Thank you for your valuable finding and recommendations made by you to improve the quality of our manuscript. We apologize for the inconsistency in the data representation and typographical representation. we have corrected and rewritten and included it in the revised manuscript (Line number 192-194; page number 9)

Comment 3. Page 9 lines 187-192. I do not understand why you think addition of 50 µl water to 50 µl naphthyl acetate and 400 µl AChE in phosphate buffer is a test of the role of AChE in ester hydrolysis. This section would make sense if the assay included no AChE, so that you would be testing the spontaneous hydrolysis of the ester.

Response to Reviewer #1 comment 3: We thank and are grateful to the reviewer for the valuable comment because he highlighted the genuine suggestion to us, to improve the quality of our manuscript. The peak of 1 naphthol was obtained in the presence of AChE , while in absence of AChE in similar condition (in water) no peak of 1 naphthol was obtained. This experiment was performed to check autohydrolysis of 1-naphthol acetate in the presence of water. We have corrected and incorporated it in the revised manuscript as per your suggestion (line number 194-196 page number 9)

Comment 4. Page 9 line 186. If you are testing spontaneous hydrolysis of naphthyl acetate, then the title of this section should be changed to “Auto hydrolysis of napththyl acetate in the absence of AChE”.

Response to Reviewer #1 comment 4:Thank you for your comment according to your suggestion we have changed the title in the revised manuscript line number 187 page no : 9

Comment 5. Page 10 line 212. Please delete “in 2.6.1”

Response to Reviewer #1 comment 5:Thank you for your comment according to your suggestion we have corrected it in the revised manuscript line number 218 page no : 10

Comment 6. Page 11 line 230. Please define a unit of activity.

Response to Reviewer #1 comment 6:Thank you for your comment according to your suggestion we have incorporated the definition of the unit of activity “The enzyme activity is measured in units which indicate the rate of the reaction catalyzed by that enzyme expressed as micromoles of substrate transformed (or product formed) per minute” in the revised manuscript line number 236-238 page no : 11

Comment 7. Page 14 line 351. The statement is incorrect that the Ellman method utilizes 10 µl of plasma sample. The original Ellman method used whole blood. The Ellman method has been adapted to assay BChE activity in plasma samples, but plasma activity was not measured by Ellman in their classic 1961 paper.

Response to Reviewer #1 comment 7:Thank you for your comment, instead of plasma we have added whole blood and incorporated it in the revised manuscript the line number 354 page number 18

Comment 8. Page 20 line 396. The statement that your HPLC assay is the best analytical approach is not convincing. You have no experience with other assays, but rely on literature reports to make your claim.

Response to Reviewer #1 comment 8: Instead of best we changed to “suitable” in line number 393 page number 20

Comment 9. Page 20 line 408. “synthetic AChE” do you mean recombinant AChE?

Response to Reviewer #1 comment 9: Yes I agree, it is Recombinant AChE and incorporated it in line number 398 and page number 20

Comment 10. The Discussion should be shortened.

Response to Reviewer #1 comment 10: We have rewritten, shorten the discussion, and removed the repeated sentences in the revised manuscript.

---

## [Editor Report · Decision Letter 3]

5 Dec 2022

A novel RP-HPLC method for quantification of cholinesterase activity in human blood: An application for assessing organophosphate and carbamate insecticide exposure

PONE-D-22-30881R3

Dear Dr. Sinha,

We’re pleased to inform you that your manuscript has been judged scientifically suitable for publication and will be formally accepted for publication once it meets all outstanding technical requirements.

Kind regards,

Oksana Lockridge, Ph.D.

Academic Editor

PLOS ONE
---

## [Editor Report · Acceptance letter]

19 Dec 2022

PONE-D-22-30881R3 

A novel RP-HPLC method for quantification of cholinesterase activity in human blood: An application for assessing organophosphate and carbamate insecticide exposure 

Dear Dr. Sinha:

I'm pleased to inform you that your manuscript has been deemed suitable for publication in PLOS ONE. Congratulations! Your manuscript is now with our production department. 

Kind regards, 

on behalf of

Dr. Oksana Lockridge 

Academic Editor

PLOS ONE